

# Quantifying large methane emissions from the Nord Stream pipeline gas leak of September 2022 using IASI satellite observations and inverse modelling

Chris Wilson[1,2], Brian J. Kerridge[3,4], Richard Siddans[3,4], David P. Moore[5,6], Lucy J. Ventress[3,4], Emily Dowd[2], Wuhu Feng[2,7], Martyn P. Chipperfield[1,2], John J. Remedios[5,6]

[1]National Centre for Earth Observation, University of Leeds, Leeds, UK.
[2]School of Earth & Environment, University of Leeds, Leeds, UK.
[3]National Centre for Earth Observation, STFC Rutherford Appleton Laboratory, Chilton, UK.
[4]Remote Sensing Group, STFC Rutherford Appleton Laboratory, Chilton, UK.
[5]National Centre for Earth Observation, University of Leicester, Leicester, UK.
[6]School of Physics and Astronomy, University of Leicester, Leicester, UK.
[7]National Centre for Atmospheric Science, University of Leeds, Leeds, UK.

*Correspondence to:* Chris Wilson (c.wilson@leeds.ac.uk)

**Abstract**

The sudden leaks from the Nord Stream gas pipelines, which began in September 2022, released a substantial amount of methane ($CH_4$) into the atmosphere. From the IASI instrument onboard EUMETSAT's MetOp-B, we document the first satellite-based retrievals of column-average $CH_4$ ($XCH_4$) that clearly show the large $CH_4$ plume emitted from the pipelines. The data displays elevations greater than 200 parts per billion (ppb, ~11%) above observed background values (1882 ± 21 ppb). Based on the IASI data, together with an integrated mass enhancement technique and formal model-based inversions applied for the first time to thermal infrared satellite methane plume data, we quantify the total mass of $CH_4$ emitted to the atmosphere during the first two days of the leaks to be 215 - 390 Gg $CH_4$. Substantial temporal heterogeneity is displayed in our model-derived flux rate, with three distinct peaks in emission rate over the first two days. Our range overlaps with other previous estimates, which were 75 – 230 Gg $CH_4$ and were mostly based on inversions that assimilated *in situ* observations from nearby tower sites. However, our derived values are generally larger than those previous results, with the differences likely due to the fact that our results are the first to use satellite-based observations of $XCH_4$ from the days following the leaks. We incorporate multiple satellite overpasses that monitored the $CH_4$ plume as it was transported across Scandinavia and the North Sea up to the evening of the 28th September 2022. We produced model simulations of the atmospheric transport of the plume using the Eulerian atmospheric transport model, TOMCAT, which show good representation of the plume location in the days following the leaks. The simulated $CH_4$ mixing ratios at three of the four nearby *in situ* measurement sites are larger than the observed *in situ* values by up to hundreds of ppb, which highlights the challenges inherent in representing short-term plume movement over a specific location using a model such as TOMCAT with a relatively coarse Eulerian grid. Our results confirm the leak of the Nord Stream pipes to clearly be the largest individual fossil fuel-related leak of $CH_4$ on record, greatly surpassing the previous largest leak (95 Gg $CH_4$) at the Aliso Canyon gas facility in California in 2015-16.



## 1. Introduction

Nord Stream is an offshore submerged pipeline network which carries natural gas from Russian facilities into Western Europe. The network is made up of two pipelines (NS1 and NS2), each originating in Russia and running through the Baltic Sea to Lubmin, Germany (Figure 1). NS1 has been operating since 2011 but the NS2 pipeline has not yet entered service, although it has carried natural gas. On 26th September 2022, multiple significant underwater gas leaks from these pipelines were detected by Nord Stream and the Danish Energy Agency, with apparently substantial gas emission through the water to the atmosphere (Danish Energy Agency, 2022). This was monitored by multiple national and international bodies over the following days. NS2 first began to leak on the morning of 26th September, from a location (15.41°E, 54.88°N) near the Danish island of Bornholm, whilst leaks were detected from NS1 at two more northerly locations (15.60°E, 55.54°N and 15.79°E, 55.56°N) later that day (Figure 1). There were reports of explosions in the area around the times that these leaks were detected (e.g. GEUS, 2022), and the pressure in the pipelines underwent an abrupt and dramatic decrease, indicative of sudden ruptures in the pipes. Neither pipeline was transporting natural gas into Europe at the time, but both contained substantial quantities of gas, the vast majority of which is methane ($CH_4$). This was released to the water and detected as large bubbles at the surface as it was further emitted into the atmosphere. Regions up to 0.7 km in diameter of rising gas bubbles were detected at the surface by *in situ* monitoring teams and by various satellite high-resolution imagers (e.g. Jia et al., 2022). The release of gas from the pipelines continued for a number of days before the Danish Energy Agency declared that the leaks had ceased on October 2nd 2022.

$CH_4$ is the second most significant greenhouse gas after carbon dioxide ($CO_2$). Human-induced emissions of $CH_4$ have been responsible for 1.19 [0.81 – 1.58] $Wm^{-2}$ of anthropogenic effective radiative forcing since 1750 (net total of 2.72 [1.96 – 3.48] $W m^{-2}$, Szopa et al. (2021)), with recent international agreements (UNFCCC, 2015; European Commission, 2021) having been put in place to urgently and significantly reduce $CH_4$ emissions for many countries. Recent satellite observations have shown that there are hundreds of $CH_4$ point source leaks worldwide contributing to direct anthropogenic emissions (e.g. Lauvaux et al., 2022). Growing levels of atmospheric $CH_4$ also adversely affect human health by contributing to increasing tropospheric ozone (West et al., 2006). A sudden large release of $CH_4$ into the atmosphere such as the one from Nord Stream could have significant consequences in terms of climate change and health. It is therefore important that the $CH_4$ emitted to the atmosphere during the Nord Stream leaks is accurately quantified. Various estimates, ranging from 75 to 230 Gg $CH_4$ (75,000 – 230,000 tonnes), have been suggested as to the quantity of $CH_4$ released to the atmosphere through assorted methodologies (see Jia et al. (2022); UNEP & IMEO (2023)).

Previous observational and modelling work (NILU, 2022; CAMS, 2022; NCEO, 2022; Jia et al., 2022) has shown that a plume of $CH_4$ originating from the location leaks was initially transported eastwards towards Finland's southern coast on 26th and 27th September, before a change in the wind direction then pushed it back out across Sweden and Norway and out into the North Sea to the north of Scotland late on the 27th and 28th. Significantly elevated near-surface $CH_4$ concentrations were briefly observed at a number of Integrated Carbon Observation System (ICOS) measurement towers in Scandinavia over the course of these three days, but there has been no direct satellite retrieval of downwind $CH_4$ concentrations available for the area to provide a more complete observation of the plume.



The Infrared Atmospheric Sounding Interferometer (IASI), on board EUMETSAT's MetOp-B satellite, is an
across-track scanning thermal infrared sounder from which $CH_4$ distributions can be retrieved twice per day
with high accuracy (Siddans et al., 2017). IASI's regular overpass times meant that it observed the area
surrounding the $CH_4$ leak at approximately 09:30 and 21:30 local time each day. Thanks to favourable
observing conditions, IASI observed enhanced $CH_4$ concentrations over the Baltic and the North Sea in the days
following the detection of the Nord Stream leaks. We use this data, together with *in situ* observations from the
ICOS network and an atmospheric chemical transport model, in order to quantify the total $CH_4$ emitted to the
atmosphere from Nord Stream during the first two days of the leaks. This is the first time that plume flux
inversions have been carried out using thermal infrared satellite data. Here we describe the results of this
quantification and put into context the derived $CH_4$ contribution from these leaks compared both with previous
similar large gas releases and with the global $CH_4$ budget.
Section 2 describes the IASI methane retrieval scheme used in this study, the $CH_4$ distributions retrieved from
the satellite and the ICOS data. Section 3 describes the atmospheric model and the inverse modelling technique.
We present our results in Section 4, before discussing their implications and concluding our discussion in
Sections 5 and 6, respectively.
**2.        Observations**
**2.1        IASI retrievals**
IASI is a cross-track-scanning Michelson interferometer (Blumstein et al., 2004) housed onboard the
EUMETSAT polar-orbiting MetOp-B satellite, which was launched in 2012. Identical instruments are hosted on
MetOp-A and -C, launched in 2006 and 2018, respectively, although MetOp-A is no longer operational. IASI
provides daily global coverage with four circular footprints of approximately 12 km diameter at nadir, arranged
in a $2 \times 2$ square grid of size $50 \times 50$ km. The IASI instrument measures upwelling thermal infrared radiation
(TIR) with 8461 channels at $0.25$ cm$^{-1}$ spectral resolution, ranging from 645 to 2760 cm$^{-1}$. Observations are
made at approximately 09:30 (descending node) and 21:30 (ascending node) local time each day. Column-
average $CH_4$ distributions used here were retrieved using an updated version (v2.0) of a scheme developed
originally for MetOp-A (Siddans et al., 2017), which has since been applied to MetOp-B (Knappett et al., 2022)
and running in near-real time at the Rutherford Appleton Laboratory (RAL) in Oxfordshire,UK
(http://rsg.rl.ac.uk/vistool). Updates included in the v2.0 scheme include improved representation of prior
covariance, changes to spectroscopy in the radiative transport model, an updated elevation model and
improvements to the representation of cloud, temperature and emissivity (Buchwitz et al., 2023). The v2.0
scheme retrieves $CH_4$ from measurements of its spectral signature in the 7.9 μm (1,260 cm$^{-1}$) region ($\nu_4$
fundamental vibration-rotation band). Vertical sensitivity generally peaks in the mid-upper troposphere since the
spectral absorption signature is determined by temperature contrast with the surface. These data have previously
been used for various studies of the atmosphere (e.g. Robson et al. (2020); Pope et al. (2021); Pimlott et al.
(2022); Buchwitz et al. (2023)).





Elevated $CH_4$ mixing ratios were observed by IASI in the Baltic Sea above the leak sites on the morning of 26th
September (Figure 2). However, cloudy conditions over much of Scandinavia and the North Sea meant that the
plume was not detected during the evening overpass on 26th September nor on the morning of 27th September.
Very high $CH_4$ concentrations were then detected over the North Sea off the west coast of Norway on the
evening of September 27th and morning and evening of September 28th. On the morning of September 28th, in
particular, a very distinct plume shape was detectable in IASI data, with areas of enhanced $CH_4$ around the
northern and southern regions of the Norwegian coast. After that day, the plume became too diffuse to be
distinguished from background concentrations. Retrieved column-averaged $CH_4$ ($XCH_4$) enhancements within
the plume on the morning of the 28th are up to 200 ppb (~11%), relative to the nearby background $CH_4$ mixing
ratios of $1882 \pm 21$ ppb (mean and standard deviation). The IASI retrievals documented here are the only
satellite observations of the Baltic Sea, Scandinavia and North Sea regions that captured a coherent $XCH_4$
plume from the Nord Stream leaks in the days immediately after the leaks began. On 30th September 2022, the
GHGSat group's satellite constellation did capture a plume as it was emitted immediately above the leak
location (GHGSat, 2022), although this was some days after the leaks began and by this point the emission rate
was fairly small (~0.08 Gg hr$^{-1}$). Although they operate at very high spatial resolution, GHGSat satellites
retrieve only the $CH_4$ enhancement above the background, rather than total $XCH_4$, and only targets specific
sources. Meanwhile, Landsat-8-OLI and Sentinel-2B also detected enhanced $CH_4$ from high resolution images
over the leak locations on 29th and 30th September (Jia et al., 2022), although these retrievals had large
uncertainties associated with them.

**2.2    ICOS network**

Consistent *in situ* monitoring of $CH_4$ mixing ratios is carried out by the Integrated Carbon Observation System
(ICOS) network (Levin et al., 2020; Heiskanen et al., 2022, https://www.icos-cp.eu/), a group of more than 140
tall tower monitoring stations located across Europe and Great Britain, including a number of measurement sites
around southern Scandinavia. These sites measure greenhouse gas mixing ratios and fluxes in the atmosphere,
ecosystems and oceans. There are four sites near Scandinavia that continuously measure $CO_2$, $CH_4$ and carbon
monoxide (CO) mixing ratios at multiple heights between 10 m and 150 m above the surface. These are located
at Birkenes, Norway (BIR, 8.3ºE, 58.4ºN, 219 metres above sea level (masl)); Hyltemossa, Sweden (HTM,
13.4ºE, 56.1ºN, 115 masl); Norunda, Sweden (NOR, 17.5ºE, 60.1ºN, 46 masl); and Utö, Finland (UTO, 21.4ºE,
59.8ºN, 8 masl). Sites are equipped with Picarro, Inc. G2401 cavity ring-down spectroscopy gas analysers,
providing continuous $CH_4$ mixing ratios with a mean difference of $0.2 \pm 0.8$ ppb compared to concurrent flask
observations (Levin et al., 2020). The sites discussed here have inlets at heights between 10m and 150m above
the ground (Hatakka et al., 2023; ICOS RI et al., 2023).

Significant enhancements of $CH_4$ (up to 490 ppb, or ~25%) were detected at each of these sites in the days
following the Nord Stream leaks (Figure 3). We compare to the highest altitude inlet for each site, which ranges
between 57m and 150m above the ground across the four sites. UTO has only one inlet height, and variations
across inlet height at BIR and NOR are less than 0.5 ppb. At HTM's highest inlet (150 masl), observed $CH_4$
mixing ratios quite different (up to 40 ppb) to the two lower inlets and we choose this inlet height to attempt to



reduce the impact of boundary layer mixing. There were relatively small CH₄ enhancements at UTO late on the
26th September, before larger enhancements were detected at NOR, HTM and finally BIR on the evening of the
next day. This distribution is consistent with the CH₄ plume from the leak being transported eastwards and then
moving back westwards across Scandinavia before it was detected by IASI off the west coast of Norway on the
27th and 28th September. Here we used the data obtained at the ICOS locations for independent verification of
our IASI-based analysis of the Nord Stream leaks.
**3.  Emission rate estimation methods and model description**
We used two methods to estimate the total mass of CH₄ in the plume observed by IASI. We first applied an
integrated mass enhancement (IME) technique, in tandem with Lagrangian model simulations, in order to
estimate the total extra mass of CH₄ contained within the plume relative to local background concentrations.
The Lagrangian model is used to inform the definition of the 'plume' and 'background' regions. This method
has the advantage that, unlike formal inversions, it is not directly dependent on the accuracy of model transport
to quantify the mass of CH₄ in the plume, but the main disadvantage is that it is not possible to exploit the
averaging kernels (AKs) of the IASI retrievals to account for the vertical sensitivity of the derived XCH₄, which
peaks in the mid-upper troposphere. It also does not account for cloudy regions in which CH₄ is not retrieved.
We therefore also employed a formal inverse modelling method based on simulation from a Eulerian chemical
transport model which allowed us to model the plume directly and to take account of the satellite AKs.
The IME methodology used the Hybrid Single Particle Lagrangian Integrated Trajectory (HYSPLIT) model
(Draxler and Hess, 1998) to produce a trajectory analysis which we combined with the XCH₄ data to determine
boundaries for the enhanced CH₄ region due to the leaks. The HYSPLIT model was initiated with GFS
meteorological data, with forward model trajectories starting at 1 km, 2km and 3km from 00:00 UTC on 26th
September, running through to 00:00 UTC on 30th September. All three trajectories showed a similar pathway
over the Baltic Sea, crossing Sweden during the morning of the 27th and reaching the Norwegian Sea by the 28th
September. These trajectories, along with the IASI observations themselves, were used to define suitable
enhanced XCH₄ regions and background regions, which represented the likely XCH₄ without the presence of the
Nord Stream plume. The background regions were defined to the west of the calculated plume trajectories, at
similar latitude ranges, away from the area affected by the leaks and over the ocean to preclude potential local
sources of CH₄. Background and enhancement regions are shown in Figure 2. The total additional CH₄ burden
was calculated by computing the difference in the mean XCH₄ concentrations over the two regions and
multiplying by the area. Estimates of the uncertainty were derived by perturbing the boundaries of the
'background' area chosen in each case with 4 scenarios, adjusting latitude- and longitude-box edges by ± 1
degree. We calculated estimates for the scenes observed on the morning of 26th September, the evening of the
27th and both the morning and evening of the 28th. The enhanced and background regions were allowed to vary
over time as the plume moved and dispersed across the North Sea. Multiple enhancement regions were
permitted within a single overpass.



We also applied an atmospheric inversion technique to the IASI data to produce an optimised time-varying
estimate of the emission rate for $CH_4$ from the leak. We used the global chemical transport model, TOMCAT
(Chipperfield, 2006; Monks et al., 2017), to simulate the emission and transport of $CH_4$ from the location of the
leak. TOMCAT has been used in a number of previous studies related to atmospheric $CH_4$ (e.g. McNorton et al.,
2016, 2018; Wilson et al., 2016, 2021; Dowd et al., 2023), along with other atmospheric species. We ran the
model at a horizontal resolution of 1.125º × 1.125º, which equates to approximately 65 km (east-west edges) ×
125 km (north-south edges) at 60ºN. There were 60 vertical levels from the surface up to 0.1 hPa. The model
dynamical time step was 5 minutes. The model was forced by meteorological data from the European Centre for
Medium-range Weather Forecasts (ECMWF) Operational analyses, regridded to the same horizontal and
vertical resolution as the model grid. The meteorological data were read into the model every 6 hours, and
linearly interpolated in time for each model time step. The initial conditions were produced from a previous
forward simulation which ran up to 00:00 26[th] September 2022. Our simulation for the inversion ran from this
time until 00:00 29[th] September 2022.

We simulated all non-plume-related $CH_4$ transport and chemistry as a separate tracer in the model, with all $CH_4$
fluxes from sources other than Nord Stream included in this background $CH_4$ tracer. Wetland emissions were
taken from the WetCHARTs inventory (Bloom et al., 2017). Anthropogenic emissions were taken from the
EDGAR v5 inventory (Crippa et al., 2020), whilst fire emissions were from GFED v4.1s (van der Werf et al.,
2017). Emissions from all other sectors, the soil sink of $CH_4$ and the monthly mean offline atmospheric loss
rates were as described in Wilson et al. (2021). Stratospheric loss rates due to $O(^1D)$ and chlorine are taken from
a previous TOMCAT full chemistry simulation (Monks et al., 2017) and hydroxyl radical distributions are based
on Spivakovsky et al. (2000). The enhanced $XCH_4$ observed by IASI is large, and the model run is short, so the
effect of uncertainties from other sources and sinks of $CH_4$ should be minimal.

The emissions from the Nord Stream leak were treated as coming from point sources in the model (at 54.88°N,
15.41°E; 55.54°N, 15.60°E; and 55.56°N, 15.79°E), although these were instantly spread across the surface
model grid cells containing the leaks. The southernmost leak was located near a model grid cell boundary in the
longitudinal direction (at 15.2°E), so this leak was split equally between the two adjacent grid cells. This
artificial instantaneous spreading out of the $CH_4$ from the leak will likely have some effect on the model's
representation of the plume movement but is unavoidable in a Eulerian model such as TOMCAT. Leak
emissions during each 3-hour time window over the simulation were tagged as separate tracers to allow for
independent scaling by the inversion (Figure S1). Figure 4 shows the TOMCAT column-averaged $CH_4$ at 08:30
UTC, the approximate IASI overpass time over the plume.

We assumed two different *a priori* (prior) emission rate distributions. The first was a constant release rate of
4.17 Gg hr$^{-1}$ (4,170 tonnes hr$^{-1}$) over the three days, emitting 300 Gg (300,000 tonnes) in total over this time.
The second distribution was an exponential decay with an e-folding lifetime of 24 hours, scaled to emit the same
total $CH_4$ over the three days. These prior emission rates are shown in Figure 5. We refer to these as the
'constant prior' and the 'decaying prior' throughout this text.



We carried out Bayesian inversions based on analytical calculation of an *a posteriori* (posterior) leak emission
rate based on finding the minimum of a cost function as in Tarantola and Valette (1982). We optimised the
mean flux from the leak locations for each 3-hour window throughout the simulation and the mean background
XCH$_4$, giving 25 optimised values in total. The mean background XCH$_4$ was given a prior uncertainty of 1%,
equal to approximately 18 ppb, and was changed very little by the inversion. All other sources and sinks were
kept unchanged. We assimilated only the data from the morning of September 28[th] (Figure 2e), since this
overpass detected the most coherent and extensive observation of the plume. We either assimilated all
observations made that morning (3980 individual retrievals, denoted 'all'), or retrievals only within the region
bounded by the longitudes 3.5°W and 9.8°E and the latitudes 58.7°N and 70.0°N, the region that contained the
main mass of the plume on the morning of 28th September (905 individual retrievals, denoted 'plume', see
Figure 6a for region definition). The AK associated with each IASI sounding was applied to the corresponding
TOMCAT methane profile. Due to the small number of variables that we optimise, and the relatively small
number of observations included, the posterior solution can be solved for directly, as has been done previously
using TOMCAT (e.g. McNorton et al., 2018; Claxton et al., 2020). See Supplementary Material and those
references for more detail of the inversion method.

We tested both the assumption that the Gaussian emissions uncertainties during each 3-hour window were
uncorrelated with each other (nocorr), and that consecutive emission windows had uncertainties with
correlations of 0.7 (corr). This value was chosen in order to impose a fairly strong correlation between emission
windows but proved to have little impact on results during emission windows that were well-constrained by
observations (See Figure 5). We tested prior uncertainties of both 100% and 50% (denoted $1.0\sigma$ and $0.5\sigma$).
Finally, instead of optimising against the full set of individual IASI retrievals, we tried optimising only the mean
XCH$_4$ value within the bounded region described above (denoted 'regional mean'). This was intended to
account for discrepancies between the simulated location of the plume compared to the observed location. In
total we therefore carried out 24 different inversions based on different prior emission distributions, sets of
assimilated data, and assumptions regarding prior uncertainties. In all inversions, the uncertainty on the
retrievals was set at 30 ppb and were assumed to be uncorrelated with each other. This value is more
conservative than the estimated individual IASI sounding uncertainty (~20 ppb), in order to attempt to account
for uncertainties from the model transport. We applied the IASI averaging kernels to represent the satellite's
vertical sensitivity in the simulated column average values. The matrices were inverted using LU decomposition
methods.

For comparison of our results with the ICOS CH$_4$ observations, we interpolate the simulated prior or posterior
mixing ratios from all tracers to the corresponding latitude, longitude and inlet heights of the ICOS sites, before
adding them together to produce simulated time series of CH$_4$ at each of the four sites. At each site, we
compared to the observational data obtained at the highest inlet height available, to attempt to reduce the
influence of boundary layer mixing.





## 4. Results

### 4.1 Integrated Mass Enhancement (IME) results

The IME method yielded various total mass estimates for each of the overpass times during the first three days of the leak. The results are shown in Table 1. The first estimate of $50 \pm 2$ Gg $CH_4$ is from an overpass that occurred only a few hours after the first leak began. Assuming that the leak commenced at 02:00 local time and that IASI was able to view most of the leaked $CH_4$ during this overpass, this implies a mean emission rate of ~6.7 Gg $hr^{-1}$ during that time. However, many nearby areas were obscured by cloud, so it is likely that IASI could not view all of the $CH_4$ emitted during these initial hours. The estimate at this time is therefore likely to be an underestimate of the total $CH_4$ release.

No plume was visible for the next 36 hours, before what was quite likely only a partial view of the plume obtained on the evening of 27th September on the west coast of Norway. The total $CH_4$ mass within this plume was $37 \pm 1$ Gg. A very clear view of the plume, which by this point was beginning to split into northern and southern sections, on the morning of 28th September yielded an inferred total of $394 \pm 9$ Gg of $CH_4$. Finally, a total enhancement of $193 \pm 6$ Gg was calculated for the evening of the 28th.

Analysis of these values is complex for two reasons. First, the effect of the IASI instrument's vertical sensitivity through application of AKs has not been taken into account. The consequences of this are hard to quantify as they depend on the vertical sensitivities of IASI both within the plume and in the background regions, and the actual vertical distribution of the $CH_4$ within the column in those regions. Using the TOMCAT model to compare the total column values in the plume with and without the AKs applied indicates that the error due to this effect may be up to 4%, although this relies on the accuracy of the model's vertical transport. Second, it is possible, and on some overpasses likely, that not all of the $CH_4$ emitted from the leak was viewed by the satellite, which would introduce a negative bias to the results.



**Table 1: Integrated mass enhancement (Gg CH$_4$) calculated from the Nord Stream plume observed by IASI over**
**three days in September 2022. Also included are the defined enhancement region and background region boundaries.**
**Overpass times with 'N/A' stated are for overpasses when the satellite's view of the CH$_4$ plume was obscured by**
**cloud.**

| Approximate local overpass time (hh:mm DD/MM/YY) | Enhancement region boundaries | Background region boundaries | Total derived CH$_4$ mass enhancement (Gg) |
|---|---|---|---|
| 09:30 26/09/22 | 53ºN – 56ºN; 13ºE – 17ºE | 64ºN – 70ºN; -4ºE – 0ºE | 50 ± 2 |
| 21:30 26/09/22 | N/A | N/A | N/A |
| 09:30 27/09/22 | N/A | N/A | N/A |
| 21:30 27/09/22 | 64ºN – 66ºN; 8ºE – 10ºE | 64ºN – 70ºN; -4ºE – 0ºE | 37 ± 1 |
| 09:30 28/09/22 | 1) 59ºN – 63ºN; -2ºE – 4.5ºE 2) 63ºN – 70ºN; 4ºE – 7ºE 3) 66ºN – 71ºN; -12ºE – -8ºE | 64ºN – 70ºN; -12ºE – -8ºE | 394 ± 9 |
| 21:30 28/09/22 | 1) 68ºN – 72ºN; -8ºE – 4.5ºE 2) 59ºN – 63ºN; 1ºE – 4ºE | 64ºN – 68ºN; -12ºE – -8ºE | 193 ± 6 |






### 4.2      Inversion results

Figure 4 shows the development of the simulated Nord Stream plume in the TOMCAT model over the first three days of the leak, assuming constant emission rates during this time. The plume initially moves northwards and eastwards during the first day. Over the following two days the plume is transported rapidly westwards across Sweden and Norway, before emerging over the North Sea at a similar time and location as indicated by the satellite observations. The plume becomes quite diffuse by the evening of 28th September.

The prior emissions, in both the 'constant' and 'decaying' configurations, underestimate the observed $XCH_4$ in the plume region on the morning of 28th September (Figure 6 and Figures S2 – S4). In addition, the simulated location of the northern section of the plume is slightly east of the observed location. This is likely due to a combination of underestimation of the initial leak rate, errors in the timing of the peak emissions in the prior and model transport errors. It is possible that the meteorological analyses used in the model and the vertical mixing parameterisation in TOMCAT combine to produce small errors in the simulated plume position. Figure 7 shows the total posterior emissions over the first two days of the leaks. In all cases, the posterior emissions are larger than the prior emissions. We report totals for only the first two days, as the observations provided by IASI on the morning of 28th September do not constrain emissions on the third day. The mean posterior emission total for these two days is $282 \pm 47$ Gg (here the reported uncertainty represents the standard deviation across the mean posterior values). The mean posterior total is $255 \pm 30$ Gg when omitting the 'regional mean' inversions where only the mean $CH_4$ value is optimised. However, there is significant variation in the posterior totals, which range between $215 \pm 13$ Gg and $374 \pm 50$ Gg, depending on the assumptions made (here the uncertainty represents the derived posterior uncertainty from the individual inversion). Total posterior emissions are consistently smaller when applying the 'decaying' prior than with the 'constant' prior, whilst posterior emissions are largest when optimising against only the regional mean, rather than against individual retrievals.

When the inversion optimises the model using the individual IASI retrievals, the position of the northern section of the plume is improved (moved further west), similar to the observations (Figure 6 and Figures S2 – S4), and simulated $XCH_4$ is increased. However, the $XCH_4$ still remains lower than the observed values. When the regional mean is optimised, the magnitudes of the simulated $XCH_4$ values are much improved, but the position of the largest values is not improved relative to the IASI observations. The remaining errors in the model representation of the plume are likely due to: i) errors in the ECMWF meteorological data, which might be improved through use of reanalyses rather than the operational analyses; ii) biases in the model transport parameterisations, particularly for vertical mixing, leading to incorrect simulated vertical distribution of the plume; and iii) uncertainties produced due to the instantaneous mixing of the leak emissions across model grid boxes.

The three-hourly posterior emission rates display significant variation over the first two days of the leaks (Figure 5). Whether the 'constant' or 'decaying' prior are used, there are three peaks in the posterior flux rates – the first during the early afternoon on the 26th September, and two more smaller peaks during the morning and afternoon of the 27th. There are low emission rates between these times. This temporal variation is consistent





across all inversions, including, to some extent, when only the regional mean XCH$_4$ is optimised (Figure S5).
The posterior emissions are far outside of the prior uncertainty during peak flux rates and, in fact, are below
zero during the night of 26$^{th}$. This negative flux is also suggestive of model transport errors. Unless temporal
error correlations are included for the prior flux in an inversion, emissions during the third day are not
constrained.

Figure 3 includes the CH$_4$ mixing ratios observed at the four ICOS sites for 26$^{th}$ – 29$^{th}$ September, and the prior
and posterior model values at those locations. The largest observed CH$_4$ enhancements above the background
concentrations were at BIR (~500 ppb), with enhancements of ~340 ppb at NOR and HTM and much smaller
enhancements of less than 60 ppb at UTO. The prior model simulations are close to the observations at UTO. At
BIR, the peaks in the prior simulations have magnitudes close to the observed value but occur around 3 hours
too early. The timing of the peak in the prior simulation at NOR is similarly early and the magnitude is 200 –
700 ppb too high. Finally, the model performance at HTM is poor, with very large simulated values, likely due
to the site's location relative to the model grid boundaries and the fast spreading of the leak emissions both
leading to excessive influence from CH$_4$ directly from the leaks. In general, the IASI-based posterior emissions
do not improve the model performance at the ICOS sites. Peak CH$_4$ at each site, which tended to be too large in
the prior simulations, has generally remained the same or increased. Posterior values at HTM have significantly
increased, whilst performance at UTO has changed little. At NOR and BIR, the posterior peaks remain too
large, although the timing of the plume reaching BIR has improved in the inversions that optimised against the
individual retrievals. At all sites, the large emissions inferred from the inversions that optimised the mean XCH$_4$
in the plume produce very large simulated mixing ratios at the ICOS sites.

**5.       Discussion**

The range of estimates from both of the methodologies that we applied to estimate the total CH$_4$ emitted from
the Nord Stream leaks using IASI retrievals of XCH$_4$ produced values greater than 200 Gg, with some estimates
reaching almost twice that value. A leak of this magnitude is by far the largest individual anthropogenic leak of
CH$_4$ to the atmosphere on record, at least twice as large as the previous largest emission event in Aliso Canyon,
California in 2015-2016 (97 Gg, Conley et al. (2016)). That leak was from a ruptured injection well pipe at a gas
storage facility near Los Angeles and continued for more than three months.

The magnitude of the Nord Stream leaks is highly significant on a global scale – when considered over a short
period. Total global CH$_4$ emissions from fossil fuels amounted to 108 Tg in the year 2017 (Saunois et al. (2020),
top-down estimate), or approximately 300 Gg day$^{-1}$. Our mean estimate from the Nord Stream leaks over two
days is therefore approximately equivalent to an extra day's emissions from global fossil fuel sources (although
it should be noted that daily emissions are likely larger today than they were in 2017). However, in the context
of annual anthropogenic CH$_4$ emissions (~364 Tg yr$^{-1}$), the Nord Stream leaks contributed only an extra 0.08%,
and  increased the annual global total CH$_4$ emissions from all sources (~600 Tg yr$^{-1}$) by just 0.05%. Chen and
Zhou (2023) calculated that a leak from Nord Stream of magnitude 220 Gg would have a negligible warming



effect on the climate ($1.8 \times 10^{-5} °C$ over a 20-year time period) and our slightly larger emission estimates would
have a correspondingly small effect.

IASI had its best view of the plume during the morning of 28th September 2022, and we base our best estimate
of the total $CH_4$ leaked to the atmosphere during the preceding two days on the observations made at that time.
Our IME method produced a value of 390 Gg $CH_4$ from those retrievals, whilst our TOMCAT inversion results
produced a range of 215 - 374 Gg, with a mean of 255 ± 30 Gg when optimising the model based on
comparisons to individual retrievals. The consistency between the results produced using the two methods is
therefore heavily dependent on the assumptions made during the inversion process, but the IME value is
approximately 50% larger than the inversion mean. This is likely due in part to the fact that the posterior
simulations still produce smaller $XCH_4$ values in the region of the plume than those observed by IASI,
indicating that the inversion-derived posterior total flux might still be too small. Indeed, the inversions that
optimise the simulated regional mean $XCH_4$ values rather than individual retrievals of $XCH_4$ produce posterior
emission totals (301 – 374 Gg) much closer to that derived by the IME method, and posterior $XCH_4$ values
similar to those observed by the satellite, albeit located too far east. In addition, the IME method does not take
account of IASI's vertical sensitivity and it is not known how results are affected by this. The effect of missing
IASI data due to cloud cover on the estimated IME value (and to a lesser extent, on the inversions) is also
difficult to quantify.

We investigated the vertical structure of the simulated plume, together with the vertical sensitivity of $XCH_4$
retrievals based on the IASI AKs (Figure S6). This shows that the northern and southern sections of the plume
during the morning of 28th September (defined as 66ºN – 71ºN, -5ºE – 6ºE and 59ºN – 63ºN, 0ºE – 7ºE,
respectively) have different vertical structures in the model. The northern section has high near-surface $CH_4$
mixing ratios from the leaks, which remain relatively constant as with altitude before decreasing until there is no
influence from Nord Stream above 500 hPa (~5.5 km). In this case, the majority of the leak-related $CH_4$ is
located beneath the peak IASI vertical sensitivity indicated by the AKs. Meanwhile, in the southern section, the
$CH_4$ contribution from the leak is smaller, but peaks higher up, at approximately 600 hPa (~4 km), around the
same region as the peak satellite sensitivity. If the vertical distributions produced in the model are correct, this
indicates that the observed $XCH_4$ in the northern and southern sections of the plume, whilst displaying similar
$XCH_4$ values, are in fact due to very different relative $CH_4$ contributions within the column. If the simulated
vertical distributions are correct, it is likely that the IME method underestimates the $CH_4$ mass in the northern
section of the plume whilst overestimating it in the southern section.

The interpretation that the inversion-derived values are low is complicated by the fact that both the prior and the
posterior simulations produce larger-than-observed $CH_4$ mixing ratios at the ICOS site locations (Figure 3). In
the model, the HTM site is located in a grid box next to the one into which the Nord Stream $CH_4$ is emitted, and
the comparison there is likely negatively and unrealistically affected by this. However, the simulated mixing
ratios at NOR and BIR are generally too large when using the prior emissions, and substantially larger when
using the posterior emissions. In fact, an inversion based only on assimilating the ICOS observations, without
the IASI data, produces a much smaller posterior total emission (88 ± 13 Gg, Figure S7). We hypothesise that



our Eulerian model's representation uncertainty is large when simulating the movement of a large distinct plume
over fixed point measurement locations, especially at the resolution used here. In addition, the model's
representation of the detailed vertical structure of the plume is key for such comparisons. The use of a high-
resolution regional model, a nested grid, or a Lagrangian model might produce better comparisons at the ICOS
sites.

Our IASI-based estimates are consistently larger than estimates produced by others using different observational
datasets. Previous estimates issued by our team and by other groups were produced quickly in the weeks
immediately following the leaks, and we have here attempted to probe the sensitivity of our results to chosen
methodologies and assumptions about the leaks and observational data. Based on ICOS observations, satellite-
based imaging spectrometer data and multiple Lagrangian models, Jia et al. (2022) calculated a total flux of 220
$\pm$ 30 Gg $CH_4$ over three days of leaks, which itself was larger than many estimates published by various groups
using a range of methods and datasets (CAMS, 2022; NILU, 2022; UNEP & IMEO, 2023). The temporal
variation of emissions produced by Jia et al. (2022) showed some similarity to our own results, with the peak
emission rate occurring during the night of $26^{th}$ -$27^{th}$ September, more than 24 hours after the leaks began. They
also computed the mass of $CH_4$ that was released from the pipelines based on pipeline dimensions and the
change in gas pressure within the pipes, calculating a value of 230 Gg. This value, along with their calculated
emission value, is smaller than the majority of our emission estimates, although a subset of our results is
consistent with their value. It remains important to investigate the roots of the apparent discrepancies between
our IASI-derived estimates and those produced via other means.

The resolution used by TOMCAT in this case (approximately $1^\circ \times 1^\circ$), is fairly coarse for capturing the
movement of the plume over the ICOS sites in particular, and results will be affected by the artificial
instantaneous spreading of the point source emissions over the comparatively large model grid cells. We can
employ Eulerian models with higher resolution, and/or Lagrangian plume models, to attempt to better represent
the plume's distribution in comparison with IASI. The effect of the meteorological data used in the models can
also be assessed through the use of reanalyses from ECMWF or other meteorological datasets. The operational
meteorological analyses used here are updated by ECMWF during reanalysis through assimilation of satellite
and *in situ* observations, which might result in better consistency between the simulated and observed plume. In
addition, investigation into the model's representation of plume uplift above the $CH_4$ release to the atmosphere
might be a key uncertainty, since it determines layer height and therefore the horizontal wind field to which the
simulated plume is exposed.

**6.     Summary and Conclusions**

We have produced the first clear satellite retrievals of column average methane that capture the $CH_4$ emitted
into the atmosphere from the Nord Stream gas leaks in late September 2022. The IASI instrument, onboard the
satellite MetOp-B, produced retrievals displaying strongly enhanced $XCH_4$ at the leak locations on the morning
of $26^{th}$ September, before large widespread enhancements were seen over the North Sea during $28^{th}$ September.



The satellite data retrieved for that day allowed us to employ two methods to quantify the CH$_4$ leaked to the
atmosphere from the Nord Stream leaks during the first two days.

Our integrated mass enhancement calculations produced total emissions of 200 – 390 Gg CH$_4$, although this
method cannot take account of the satellite instrument's vertical sensitivity, which peaks in the mid-upper
troposphere, and cannot account for regions of enhanced CH$_4$ that are not observed due to clouds. We also used
formal Bayesian inversion methods, using the TOMCAT atmospheric chemical transport model, to quantify the
emissions based on the observations made on the morning of 28$^{th}$ September. This is the first time that plume
flux inversions have been carried out using thermal infrared satellite data. Here, we investigated the effect of a
range of assumptions within the inversion, including the prior distribution of the emissions, the related prior
uncertainties and the way that observations are assimilated. We calculated total emissions between 215 and 374
Gg. The mean over all inversions is approximately $282 \pm 47$ Gg, whilst the mean over the inversions that
optimise against individual IASI retrievals is $255 \pm 30$ Gg. All of our results imply that the Nord Stream leaks
were by far the largest recorded individual anthropogenic leak of CH$_4$ to the atmosphere.

Our estimates are larger than previous values given for the Nord Stream leaks, produced using alternative
observational data. There are large differences between our posterior results and *in situ* observations made in the
region, and more work is necessary to discern to what extent this is due to errors in the flux estimates produced
from the satellite data and how much is due to poor model plume representation at the tall tower locations. Our
ability to monitor, simulate and quantify leaks of GHGs and pollution events such as this one is continuously
improving, aiding our ability to mitigate the human influence on the atmosphere. It is also clear from this study
that thermal infrared instruments such as IASI, which have peak sensitivity high in the troposphere, are able to
provide more information concerning surface events such as the Nord Stream leaks than might have been
appreciated previously. In any case, whilst this particular event remains highly significant locally over a short
time period, the effect of these emissions, by themselves, is very small in terms of both the global atmospheric
CH$_4$ budget and the climate.

**Data Availability**

MetOp-B IASI methane observations up to March 2021 are available on the Centre for Environmental Data
Analysis (CEDA) long-term data archive (Knappett et al., 2022). More recent data, including the near-real time
(NRT) data for the period covering the Nord Stream leaks, is viewable through the public visualisation tool
(http://rsg.rl.ac.uk/vistool, last access 18/07/2023). NRT data is available through contacting the authors. The
TOMCAT model output for this period will be made available on the Centre for Environmental Data Analysis
(CEDA) long-term data archive upon publication of this work. The ICOS methane concentrations were
downloaded from ICOS Carbon portal (https://data.icos-cp.eu/portal/, last access 18/07/2023).






**Author contribution**

CW, BJK, and JJR conceptualised the study. BJK, RS, and LJV produced the satellite data. CJW, DPM, ED, WF and MPC carried out data analysis and modelling. All co-authors contributed to the design of the study and to writing the manuscript.

**Competing interests**

The authors declare that they have no conflict of interest.

**Acknowledgements**

This work was funded by the Natural Environment Research Council through its grants to the UK National Centre for Earth Observation (NCEO; NERC grant numbers NE/R016518/1 and NE/N018079/1). The IASI retrievals were produced using JASMIN, the UK collaborative data analysis facility, at the Rutherford Appleton Laboratory. The TOMCAT model simulations were carried out using ARC4, part of the High-Performance Computing facilities at the University of Leeds, UK. EUMETSAT provided data for MetOp-B IASI, MHS & AMSU-A data and ECMWF provided meteorological data used in NRT processing system and TOMCAT simulations. We thank ICOS PIs for providing their methane concentration data.



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



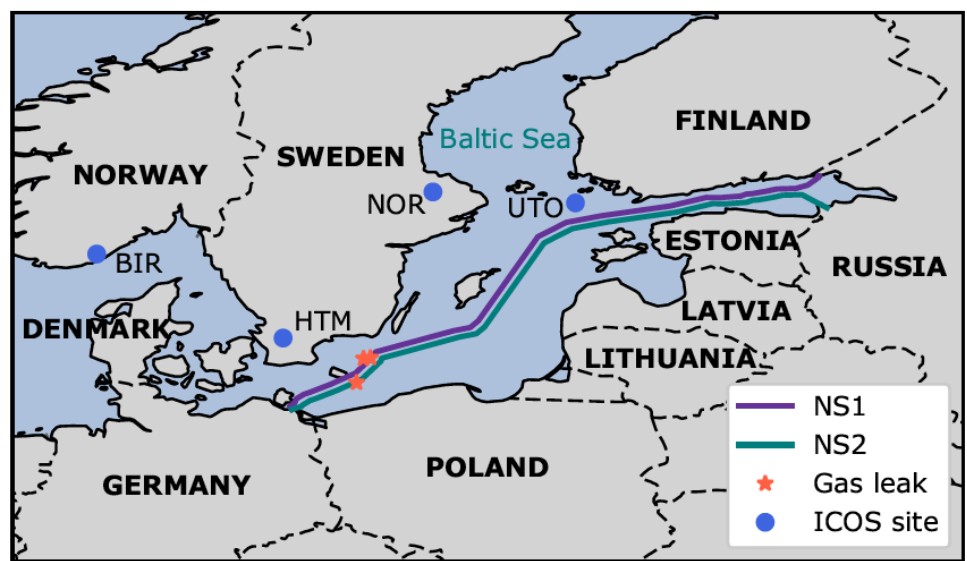

**Figure 1: Map showing Nord Stream pipeline routes (teal and purple lines), gas leak locations (red stars) and** *in situ*
**ICOS monitoring site locations (blue circles).**

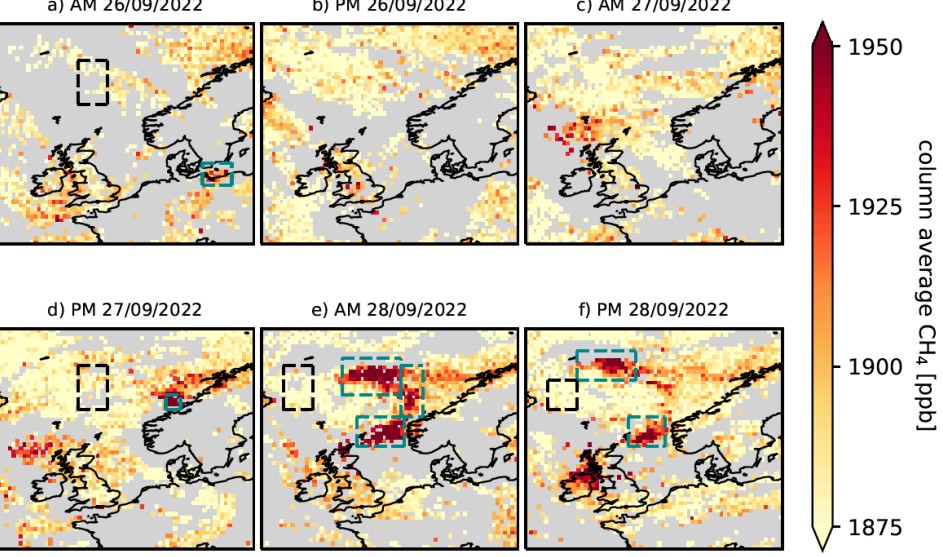

**Figure 2: IASI column average CH$_4$ (ppb) for 26th - 28th September 2022. Retrievals are averaged onto 0.25º × 0.25º**
**grid boxes, weighted inversely to their uncertainties for the morning and evening overpasses of each day. Black**
**dashed boxes show 'background' regions used in the integrated mass enhancement (IME) method, whilst turquoise**
**dashed boxes show 'enhancement' regions. Grey regions are obscured by cloud.**



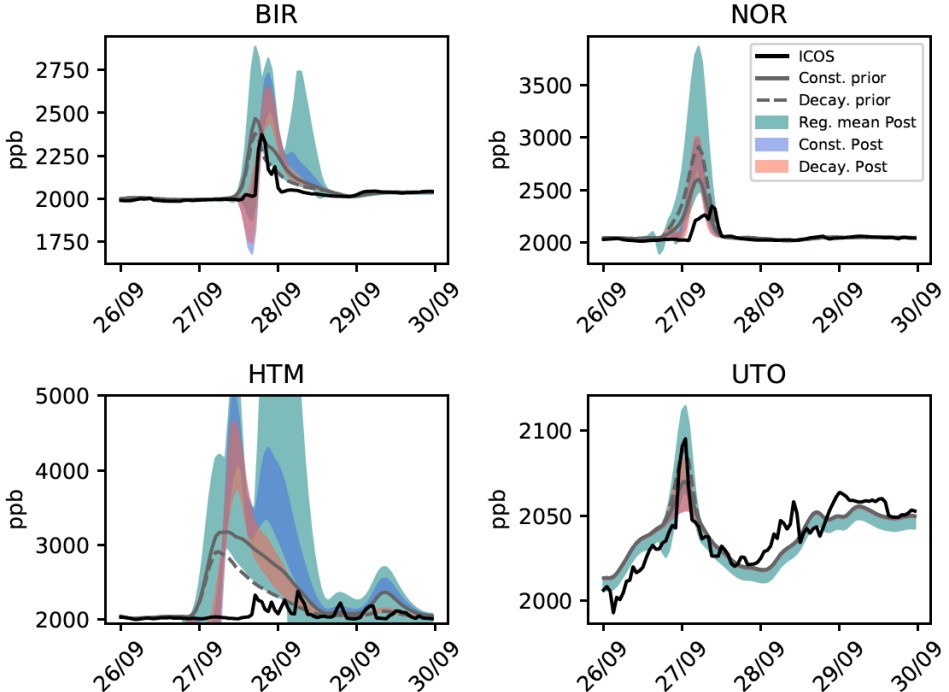


Figure 3: Observed (black line) and simulated (grey lines/coloured shading) CH$_4$ mixing ratios (ppb) at Integrated
Carbon Observation System (ICOS) sites during 26$^{th}$ – 29$^{th}$ September 2022. Observations and model output are both
averaged into hourly means. ICOS sites are at Birkenes, Norway (BIR), Norunda, Sweden (NOR), Hyltemossa,
Sweden (HTM) and Utö, Finland (UTO). See main text and Figure 1 for further details. Grey lines show TOMCAT-
simulated CH$_4$ using the two prior emission estimates, and shaded regions show the simulated min/max range for the
inversions with constant prior (blue) and decaying prior (red) optimised against individual retrievals, and for
inversions optimised against the regional mean (teal). Inlet heights are the highest available at each site: 75m at BIR;
100m at NOR; 150m at HTM and 57m at UTO. Note the different y-axis ranges in each panel.

718
719



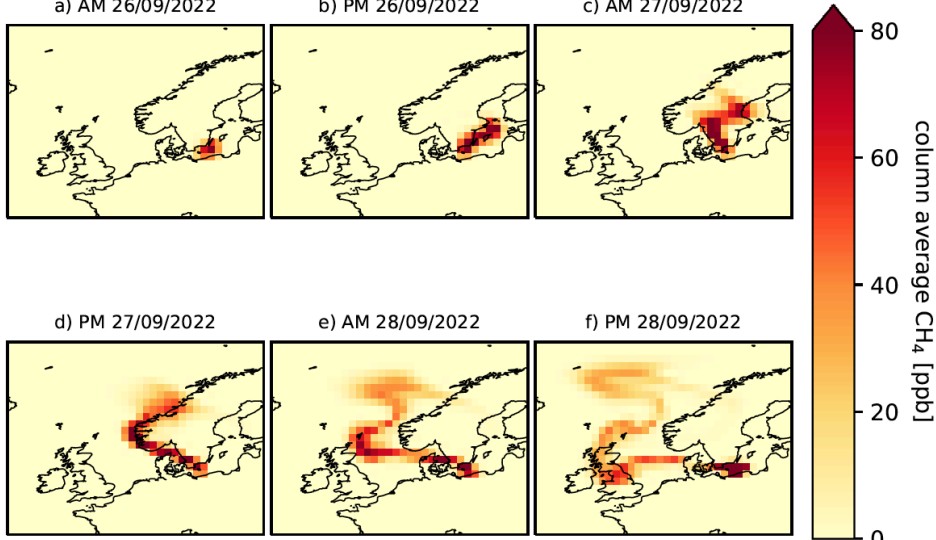

**Figure 4: Simulated TOMCAT column average CH$_4$ (ppb) from Nord Stream gas leaks for 26$^{th}$ – 28$^{th}$ September 2022. Background CH$_4$ and emissions from sources other than Nord Stream are not included. Output times are matched to IASI local overpass times, but IASI averaging kernels have not been applied. Column averages are displayed on the model grid with horizontal resolution 1.125º × 1.125º. Emission rates from the leaks is constant at 4.17 Gg hr$^{-1}$, summing to 300 Gg in total over the three days.**



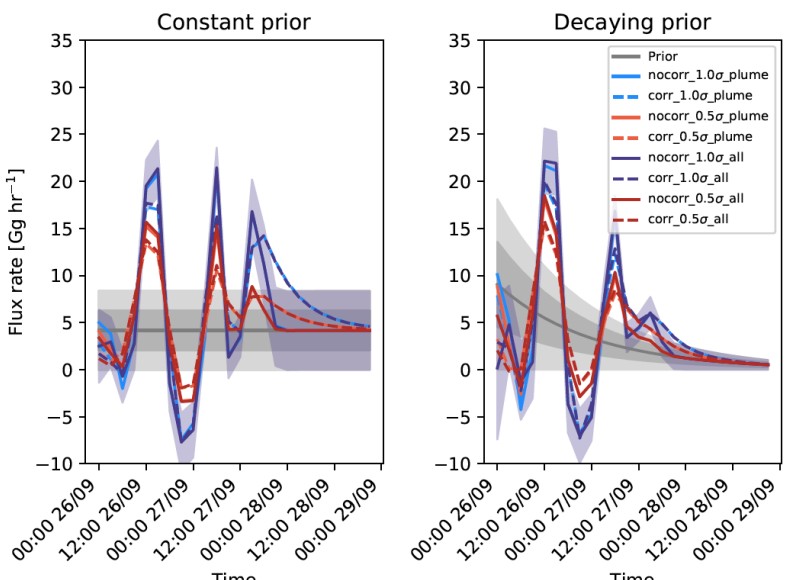

**Figure 5: Prior and posterior CH$_4$ flux rates (Gg hr$^{-1}$) over the first three days (September 26th - 28th) of the Nord Stream leaks based on IASI data from the morning of 28$^{th}$ September 2022. Prior flux rate is shown in grey, with dark grey shaded region showing the 50% prior uncertainty and the light grey shaded region showing the 100% prior uncertainty. Dashed lines show posterior inversions with prior temporal correlations imposed; solid lines show those without prior correlations. Blue lines show inversions with 100% prior uncertainty imposed; red lines show those with 50% prior uncertainty. Darker shades show inversions based on all available IASI data; lighter shades show inversions based only on IASI data from near the plume, in the region highlighted in Figure 6. Shaded blue region shows the posterior uncertainty for the 'nocorr_1.0σ_all' case.**




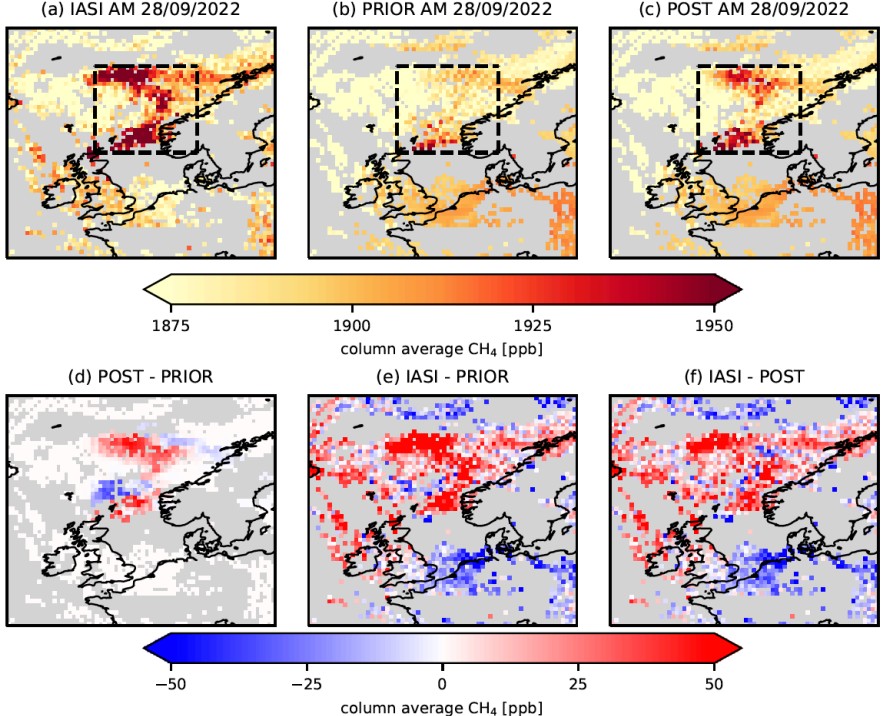

**Figure 6: Column average CH₄ (ppb) on the morning of 28ᵗʰ September over the region of the Nord Stream gas leaks from (a) IASI; (b) TOMCAT using the constant prior emissions; and (c) TOMCAT using the nocorr_1.0_plume posterior emissions based on that prior. Also shown is the difference between the model posterior and prior (d); the difference between IASI and the model prior (e); and the difference between IASI and the model posterior (f). Retrievals and model output are averaged onto 0.25° × 0.25° grid boxes, weighted inversely to the observations' uncertainties. IASI averaging kernels are applied to the TOMCAT output. Black dashed line shows the 'plume' region defined in the text, used for optimising only the regional mean XCH₄ value.**




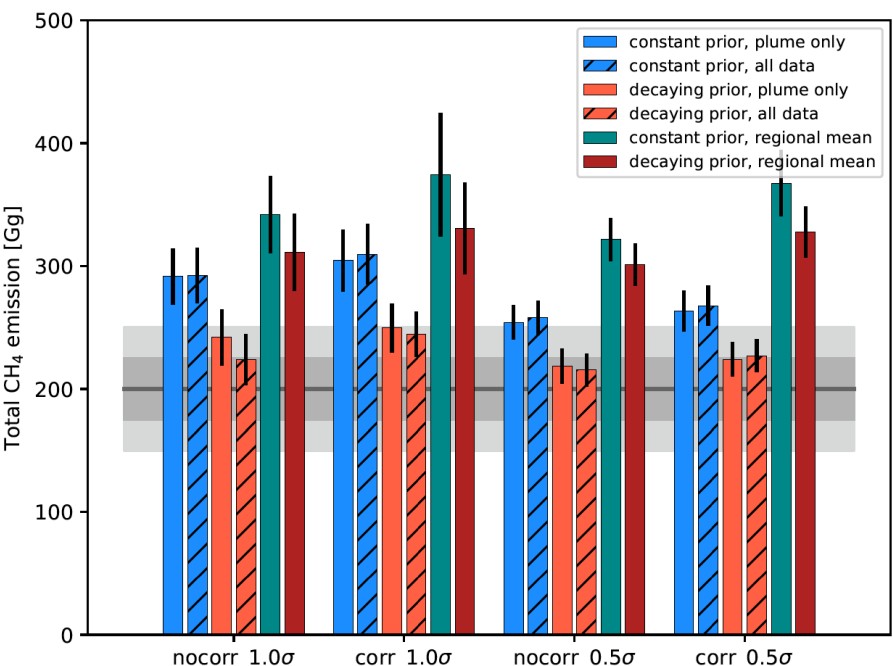

Figure 7: Total (two-day) posterior CH₄ emissions (Gg) from the Nord Stream leaks during 26th – 27th September
based on multiple different IASI-based inverse modelling calculations. Blue bars represent inversions with the
constant prior where the model is optimised against individual IASI retrievals, whilst orange bars are the same but
for the decaying prior. Turquoise bars represent inversions with the constant prior where the model is optimised
against the mean XCH₄ in the plume region, whereas red bars are the same but for the decaying prior. Hatched bars
show inversions in which all IASI data is included, and unhatched bars show inversions in which retrievals only
within the plume region are included. 'Corr' and 'nocorr' refers to inversions with and without prior temporal
correlations included, whilst 1σ and 0.5σ refer to inversions with 100% and 50% prior uncertainty. The grey solid
line shows the prior emission total, with 50% and 100% 3-hour prior uncertainty shaded in dark and light grey,
respectively.