# Peer review of "Quantifying large methane emissions from the Nord Stream pipeline gas leak of September 2022 using IASI satellite observations and inverse modelling"

_EGUsphere, 2023_

## Referee Comment (RC2)

This manuscript uses the IASI retrievals of XCH4 over North Sea to estimate the methane emissions due to the leakage from Nord Stream pipelines in September 2022. The authors use two ways, the Integrated Mass Enhancement (IME) and a Bayesian inversion based on a 3-D transport model TOCAT, to calculate the emissions during 26th to 28th, September 2022. This is the first study to use the satellite methane retrievals to estimate the emissions from this event. The scope of the paper fits ACP well. However, there are many aspects need to be fully addressed before it can be published on ACP.

General comments:

1, IASI retrievals are the only satellite observations that recorded coherent XCH4 enhancement during the leakage from Nord Stream pipeline. As mentioned by authors, the plume initially transported eastwards then westwards, but only IASI observations over the North Sea from 26-28th were shown in the paper. Does IASI also document the similar plume enhancements over Baltic Sea? If not, it is kindly suggested to adjust the relevant statement that IASI is the only satellite retrieval that documents the entire event.

2, The role or motivation of using HYSPLIT model is not clear. If I understand correctly, HYSPLIT model is used to figure out trajectories of the leakages then determine the regional background. However, the regional background will not change too much during a short period as in this case. Using a temporal average over a certain region to be the background does not affect the result of IME method too much. Then HYSPLIT model is not needed.

Another option is to use the simulation of HYSPLIT model to complete the "plume shape" that are partly covered by cloud. The incomplete IASI observations can inevitably cause a large uncertainty when using the IME method. If the trajectory of HYSPLIT can be better used, it firstly can help to validate the evolution of the plume on 28th (i.e., separate into southern and norther parts). Secondly, the emission estimates of IME can be more comparable with the results from TOMCAT.

3, When using TOMCAT to do a reversion, the simulations at four sites with a priori emission rates show large discrepancies to in-situ observations (Figure 3). To me, it is not clear that why a priori emission rate is a fixed constant as 4.17 Gg hr$^{-1}$? Did the authors test other numbers that can derive more reasonable results for the comparisons between the simulation and in-situ observations?

Apart from site HTM, variations of simulations with a priori emission rate at other three sites look reasonable considering the coarse resolution of TOMCAT. It also implies that the metrological data and dynamics used in TOMCAT simulations are not main reasons cause large discrepancies. The coarse resolution, as mentioned in the paper, can be one of the important reasons. However, Fig. 6 indicates the negative biases are also very large over land comparing to IASI. Thus, the coarse resolution is not the key reason for

such big discrepancies between TOMCAT and IASI.

4. The results including the IASI observations even became much worse. Followed by the above comment, it would be helpful if the authors can compare simulations with satellite observations at some background sites before the analysis. On one hand, this comparison can be used to check if the system bias exists between model and IASI. On another hand, the satellite observations can be quite noisy at high latitudes. It also helps to assess the quality of the observations over North Sea.

To a regional revision by using a transport model like TOMCAT, the boundary condition and initial condition can be quite important. There is little discussion about them, even the spatial distribution of the emission inventory used in this study is not presented and discussed. It is quite difficult to understand why the difference between model simulations with a priori and a posterior inventory only occurs in the dashed black box (plume region) showing in Fig. 6d.

Specific comments:

1, Line 54: It would be better to refer two gas leak points marked with longitude and latitude to red asterisks (i.e., Gas leak 1, 2, 3) in Figure 1.

2, Line 124: It would be better to add longitude and latitude along figures. The names of countries are also suggested to be added in Figure 2. The readability can be improved in this way.

3, Line 132-134: If I understand correctly, the authors mainly show temporal variations of methane concentrations over North Sea. Are there any observations of IASI over Baltic Sea, where the leakage occurred? If not, the authors should clarify the lack of satellite observations, although IASI retrievals are the only satellite observations that captured a coherent $XCH_4$ enhancement in the days immediately after the leaks began.

4, Line 193-196: The areas of background in Figure 2 are decided by HYSPLIT model and the uncertainty is estimated by perturbate the area of a black box. However, the number of available observations in each black box is different. How significant is the impact of sampling bias?

5, Some $XCH_4$ enhancements are quite strong near the western coast of Norway and northern coast of the UK even before 28[th]. What can be the possible reasons for this?

6, Line 236: Why is this emission rate selected? Any background information or explanation?

---

## Author Comment (AC1)

**Reviewer 1**

**General comments**

This manuscript uses total column methane mixing ratios (XCH4) from the IASI instrument onboard the MetOp-B satellite to estimate the amount of CH4 released from the Nordstream pipeline leaks in September 2022. There have been previous studies that have attempted to estimate the source from ground-based observations, but this appears to be the first study using satellite retrievals for the source estimate. Based on two methods, the Integrated Mass Enhancement (IME) approach and a Bayesian inversion using a Eulerian atmospheric transport model, the authors find a source of 215-390 Gg for the first 2 days following the first pipe rupture. The lower end of this range overlaps with bottom-up and other observation-based estimates.

This study provides another constraint on this enormous (but also short-lived) CH4 source and is a nice example of how satellite observations can generally provide constraints on CH4 emissions. On the other hand, there are a number of shortcomings which I think need to be addressed before being accepted for publication in ACP.

We thank the reviewer for their comments, which have significantly helped to improve the paper and clarify our results and message. We hope that we have addressed these concerns appropriately. Our point-by-point response is given below, highlighted as blue text.

It should be noted that whilst updating the IME values based on comments made here by both reviewers, an error was discovered in the code that was used to derive the IME flux values. The error was related to the calculation of grid cell area. Fixing this problem has had the effect of reducing the IME-derived flux values by 50% or more at each overpass time. The most complete estimate, produced using the overpass on the morning of September 28th 2022, has been reduced to approximately 161 Gg CH4, from 394 Gg CH4. We have updated the manuscript with the new values and adapted the text to discuss the comparison between the inversion and IME values.

The fact that the corrected IME value is now lower than the inversion-derived value is consistent with the fact that the satellite averaging kernels (AKs) are not taken account of using the IME method. IASI's vertical sensitivity is in the middle and upper troposphere, away from the level of the peak CH4 values (according to the TOMCAT simulation). This fact should lead to an underestimation of the true flux value when using the IME method, and so the updated results can be consistent with the model-derived values.

The first is that the IME method, at least in the way it is used here, does not really seem suitable to the problem since the plume is not well-defined and is partially obscured by cloud. However, it could be interesting to combine this method with the Lagrangian model

(HYSPLIT) simulations in a point-source inversion. This would entail using HYSPLIT to model the plume using a prior estimate of the source, and then the IME could be compared to the modelled IME for areas of the plume identified (e.g. Fig. 2) and then the difference between these estimates could be used to optimize the source. In this way, the problem of cloud could be circumvented, and a more complete estimate of the source could be obtained. It should be more comparable then to the estimates obtained using the Eulerian transport model, TOMCAT.

We understand the thinking put forward by the reviewer here and thank them for this comment. Our intention for the use of the IME method here was for it to be an empirically-driven estimate of the Nord Stream emission, based only on the observed XCH4 concentrations, rather than a formal inversion using of a model. The intention in this case is to reduce the potentially large impact of model transport error. We acknowledge that there are drawbacks in this philosophy and we attempt to describe these thoroughly in the text. Although we used HYSPLIT here in a limited manner, to ensure that our defined plume and background regions were respectively related or not related to the Nord Stream leak, we are reluctant to employ HYSPLIT within a formal inversion, for which we are already using the TOMCAT model.

The drawbacks inherent in this choice include the fact that on many overpasses, some parts of the plume were obscured by cloud, and that the satellite averaging kernels could not be taken account of. We discuss these disadvantages in the main text of the paper. In terms of the cloud cover problem, which the reviewer mentions here as a major issue, we agree that on some days, the estimates appear to be significantly affected by cloud. However, the clear view that IASI obtained of the plume on the morning of September 28[th] was almost entirely unaffected by cloud cover, as confirmed by the TOMCAT simulations. For both the TOMCAT inversions and the IME estimates, this is the view on which we base our main conclusions. We do not attempt to draw conclusions from the IME results on any of the other days. We have clarified this in the main text as follows:

"It is clear that for the majority of the IASI overpasses, at least some part of the plume is unfortunately obscured by cloud. The only clear view of the plume is obtained on the morning of the 28th September, as confirmed by the TOMCAT simulations. We therefore suggest that the IME-related estimates from the other overpasses are likely to be underestimates of the total CH4 released by Nord Stream, and we base any conclusions on the estimate obtained on the morning of September 28th."

The second is that a rather poor estimate of the prior emissions from the pipelines is used. It would not require much effort to come up with a physical model for the emissions, which would give a more accurate representation of the amount of CH4 released and of the release rate. I do not understand why this was not done.

We agree with this point and have therefore included an extra prior distribution in the paper, along with the two that were previously included. Instead of producing our own physical model of the release rate, we used the results from Poursanidis et al. (2024), who had their own model of the release. This model shows a temporal variation more in line with that suggested as most likely by this reviewer. We have included extra results and discussion in the paper related to this additional experiment. This has led to significant changes in the main text and the replacement of some of the figures, so we will not detail all changes here. The inclusion of this prior has not affected our main findings, although it has revised our mean estimate for the Nord Stream leak upwards by approximately 10%.

The third is that errors in the representation of atmospheric transport by the relatively coarse TOMCAT model are not really addressed. The authors mention this as a problem but do not attempt to reduce the effect of these errors. The authors perform one inversion where the mean of enhanced CH4 is used rather than the individual retrievals, which gave a rather different result for the emissions. I suggest the authors explore an intermediary approach using averaged retrievals, where the averaging is sufficient to reduce errors due to incorrect positioning of the plume, but where there is still some spatial information.

We agree that this is a useful intermediate step and have now included results from a set of inversions using observations averaged into 3° × 3° grid cells. We chose this resolution to be coarser than the model resolution, to attempt to reduce the effect of errors in the simulated plume location. Results from these inversions are included in the Supplementary materials and discussed in the main text in Section 4.2.

Also, I think the authors should examine more carefully why there is a big discrepancy between the modelled and observed mixing ratios at three of the four ICOS sites, and check if the most recent ICOS data were used, since the first data submission missed some of the peaks (see specific comments below).

We thank the reviewer for this suggestion. They were correct to query the version of the ICOS data that we used for comparison. Please see specific response below.

**Specific comments**

L20: Suggest authors specify the date of the first explosion rather than just "September 2022" since in L33 the author's state they incorporate data up to 28 September, so for reader's it would be interesting to know how many days this is after the start of the leaks.

Agreed. We have now added this information on the first line of the abstract.

L146: I think there are only 46 atmospheric stations in the ICOS network (see: https://www.icos-cp.eu/observations/atmosphere/stations). If the authors include flux sites in the 140 stations that they mention, this is inconsistent with the fact that they label these as "tall tower monitoring stations", also, the flux sites are not relevant to this study.

It is correct that there are 46 atmospheric stations in the ICOS network, along with a number of flux sites, and our statement in the text was incorrect. We have now clarified this as follows:

"a group of more than 140 monitoring sites located across Europe and Great Britain, including a number of measurement sites around southern Scandinavia. These sites measure greenhouse gas mixing ratios and fluxes in the atmosphere, ecosystems and oceans. The network includes 46 tall tower sites across 16 countries that measure greenhouse gas concentrations in the atmosphere, along with meteorological parameters. These include four sites near Scandinavia… "

L158: In the first submission of ICOS data, some high values of CH4 associated with the Nordstream leaks were filtered out in the automatic quality control. This was corrected in a subsequent submission. Could the authors please confirm which version of ICOS data they used, and if this version was the corrected one?

This is a very important point – we were in fact using the 'near real time' version of the ICOS data. That version of the data had filtered out some of the highest mixing ratio values, particularly at the BIR site. Thanks to this recommendation, we have updated to the finalised version of the ICOS data. We have also updated all instances of usage and reference to this data in the paper's text and figures. We thank the reviewer for pointing this out as it significantly affects some of the conclusions of the paper, regarding the comparison between the (IASI-assimilated) posterior model output and the ICOS data.

Now, the posterior comparison between the model and the observations at BIR are much improved compared to the prior. However, both the prior and the posterior versions of the model continue to overestimate the $CH_4$ spike observed at the NOR site. There have been substantial changes in the manuscript relating to this update, including an updated version of Figure 3 and updated discussion in the paragraphs starting on lines 363 and 429 (line numbers refer to the preprint version of the text).

L194-195: The grid cells do not all have exactly the same area, so do the authors take the area-weighted average for the calculating the additional CH4 burden?

We were not previously area-weighting the mean XCH4 value, but we have updated our calculations to include this. We have updated Table 1 and the main text of the paper, but please see our earlier comment regarding other changes to these values due to correcting our methodology.

L236-238: Why was a constant release rate used as a prior estimate when very clearly the release was not constant. The first pipe rupture occurred early on the morning of 26/09, while the subsequent 3 ruptures occurred in the evening of 26/09, so this information should be reflected in the prior. The calculation of the second prior release rate also seems rather primitive, why not use a physical calculation of the release based on the length and inner diameter of the pipes and the initial pressure, and combined with what is known about the timing of the ruptures?

As mentioned in the 'general comments' section above, this is an important point and we have now included a more realistic representation of the release rate, based on the criteria suggested by the reviewer. We have now used, in addition to our other priors, the values derived by Poursanidis et al. (2024), and have updated all figures and text to reflect this.

L246-247: If I understand correctly, only the Nordstream leak source was optimized. This assumes that the estimates for the non-leak sources are accurate. What is the uncertainty from this assumption? Could the authors estimate this uncertainty by trying different prior source estimates? Or include these sources also in the optimization?

We accounted for uncertainty concerning $CH_4$ sources not related to Nord Stream by including the background XCH4 concentrations in the inversion. Optimising the non-Nord Stream emission inventory will likely have a smaller effect than optimising background $CH_4$ directly. As already discussed in the main text the effect of uncertainty related to emissions relative to the elevated XCH4 values observed in the Nord Stream plume should be small. We have included a new figure displaying the emission inventory used in the model in the Supplementary material, and we have clarified the effects of the other sources in the Discussion section as follows:

"The uncertainty induced by the emission inventories should be small compared to the observed plume-related concentrations during a short simulation such as this one, but the initial conditions could introduce biases between the model and satellite. We attempt to account for this through inclusion of the background XCH4 in the inversion state vector, but further investigation into the effect of the initial conditions is warranted."

L247-248: Only one overpass on morning of 28/09 was assimilated, so there is little constraint actually on the temporal evolution of the leak source, thus, it would be even more important to have a good temporal evolution in the prior estimate.

Agreed. As stated above, we have now included a more realistic prior release rate in our results.

L266: It's not clear to me what these 24 different inversions are. Could these be summarized in a table?

Yes. We have now included a table describing all of the inversions in the Supplement, as Table S2.

L295: Most of the plume must have been obscured on 27/09, the actual CH4 enhancement should not have been so much lower compared to that on 26/09 and 28/09.

Yes, correct. We were trying to make this point in the text, but we have now stated it more explicitly, as follows:

"The total CH4 mass within the small observed section of this plume was 37 ± 1 Gg, the low value likely due to much of the plume being obscured by cloud."

L363: Same comment as for L158, the authors should check which version of the ICOS data they used.

Yes, as stated above we have now updated the ICOS data used in the paper.

L408: I think the authors could explore a bit further the method of comparing modelled mean XCH4 values over given regions with observed means, which would be perhaps a way of reducing the impact of the transport uncertainty, which appears to be quite significant (but not altogether unexpected considering the resolution of the model and that the source is only in the order of 100s meters in diameter). Perhaps this could be done by comparing the model with averages of retrievals?

As stated above, we have now included averages of retrievals in our results. See response in 'General comments' for more details.

L450: A peak emission rate in the night of 26-27/09 is actually to be expected because the subsequent 3 ruptures of the pipeline occurred at around 19:00 local time on 26/09. Thus it is only a few hours after 3 of the 4 leaks started.

A good point. We have updated in the text here, as follows:

"...with the peak emission rate occurring during the night of 26th -27th September, more than 24 hours after the first of the leaks began, but only a few hours after three of the four leaks started."

**Reviewer 2**

This manuscript uses the IASI retrievals of XCH4 over North Sea to estimate the methane emissions due to the leakage from Nord Stream pipelines in September 2022. The authors use two ways, the Integrated Mass Enhancement (IME) and a Bayesian inversion based on a 3-D transport model TOCAT, to calculate the emissions during 26th to 28th, September 2022. This is the first study to use the satellite methane retrievals to estimate the emissions

from this event. The scope of the paper fits ACP well. However, there are many aspects need to be fully addressed before it can be published on ACP.

We thank the reviewer for their comments, which have significantly helped to improve the paper and clarify our results and message. We hope that we have addressed these concerns appropriately. Our point-by-point response is given below, highlighted as blue text.

It should be noted that whilst updating the IME values based on comments made here by both reviewers, an error was discovered in the code that was used to derive the IME flux values. The error was related to the calculation of grid cell area. Fixing this problem has had the effect of reducing the IME-derived flux values by 50% or more at each overpass time. The most complete estimate, produced using the overpass on the morning of September 28th 2022, has been reduced to approximately 161 Gg CH4, from 394 Gg CH4. We have updated the manuscript with the new values and adapted the text to discuss the comparison between the inversion and IME values.

The fact that the corrected IME value is now lower than the inversion-derived value is consistent with the fact that the satellite averaging kernels (AKs) are not taken account of using the IME method. IASI's vertical sensitivity is in the middle and upper troposphere, away from the level of the peak CH4 values (according to the TOMCAT simulation). This fact should lead to an underestimation of the true flux value when using the IME method, and so the updated results can be consistent with the model-derived values.

**General comments:**

1, IASI retrievals are the only satellite observations that recorded coherent XCH4 enhancement during the leakage from Nord Stream pipeline. As mentioned by authors, the plume initially transported eastwards then westwards, but only IASI observations over the North Sea from 26-28th were shown in the paper. Does IASI also document the similar plume enhancements over Baltic Sea? If not, it is kindly suggested to adjust the relevant statement that IASI is the only satellite retrieval that documents the entire event.

Agreed. IASI was able to retrieve over the Baltic Sea near the source location only on the morning of September 26[th], after which the region was obscured by cloud. To improve clarity, we have adapted this sentence on line 132 to read as follows:

"The IASI retrievals documented here are the only satellite observations that captured a coherent XCH4 plume from the Nord Stream leaks over the North Sea in the days immediately after the leaks began."

2, The role or motivation of using HYSPLIT model is not clear. If I understand correctly, HYSPLIT model is used to figure out trajectories of the leakages then determine the regional background. However, the regional background will not change too much during a short period as in this case. Using a temporal average over a certain region to be the background does not affect the result of IME method too much. Then HYSPLIT model is not needed.

Another option is to use the simulation of HYSPLIT model to complete the "plume shape" that are partly covered by cloud. The incomplete IASI observations can inevitably cause a large uncertainty when using the IME method. If the trajectory of HYSPLIT can be better used, it firstly can help to validate the evolution of the plume on 28th (i.e., separate into southern and norther parts). Secondly, the emission estimates of IME can be more comparable with the results from TOMCAT.

We understand the thinking put forward by the reviewer here and thank them for this comment. Our intention for the use of the IME method here was for it to be an empirically-driven estimate of the Nord Stream emission, based only on the observed XCH4 concentrations, rather than a formal inversion using of a model. The intention in this case is to reduce the potentially large impact of model transport error. We acknowledge that there are drawbacks in this philosophy and attempt to describe these thoroughly in the text. Although we used HYSPLIT here in a limited manner, to ensure that our defined plume and background regions were related, or not related, to the Nord Stream leak, we are reluctant to employ HYSPLIT within a formal inversion, for which we are already using the TOMCAT model.

The drawbacks inherent in this choice include the fact that on many overpasses, some parts of the plume were obscured by cloud, and that the satellite averaging kernels could not be taken account of. We discuss these disadvantages in the main text of the paper. In terms of the cloud cover problem, which the reviewer mentions here as a major issue, we agree that on some days, the estimates appear to be significantly affected by cloud. However, the clear view that IASI obtained of the plume on the morning of September 28[th] was almost entirely unaffected by cloud cover, as confirmed by the TOMCAT simulations. For both the TOMCAT inversions and the IME estimates, this is the view on which we base our main conclusions. We do not attempt to draw conclusions from the IME results on any of the other days. We have clarified this in the main text as follows:

"It is clear that for the majority of the IASI overpasses, at least some part of the plume is unfortunately obscured by cloud. The only clear view of the plume is obtained on the morning of the 28th September, as confirmed by the TOMCAT simulations. We therefore suggest that the IME-related estimates from the other overpasses are likely to be underestimates of the total CH4 released by Nord Stream, and we base any conclusions on the estimate obtained on the morning of September 28th."

3, When using TOMCAT to do a reversion, the simulations at four sites with a priori emission rates show large discrepancies to in-situ observations (Figure 3). To me, it is not clear that why a priori emission rate is a fixed constant as 4.17 Gg hr-1 ? Did the authors test other numbers that can derive more reasonable results for the comparisons between the simulation and in-situ observations?

We chose this rate as it gives a total emission value of 200 Gg CH4 over the first two days of the period assessed in this study. This value was chosen to be in line with results from other publications (e.g. Jia et al. (2022)) and our own previous simulations (https://www.nceo.ac.uk/article/220000-tonnes-of-methane-likely-released-from-nord-stream-gas-leak/). We had tried alternative (smaller and larger) prior release rates values in our previous work, but settled on 200 Gg over the initial two-day period as a compromise that produced relatively good comparisons at the tower locations but provided flexibility for the IASI-based inversion. We have clarified this in the text as follows:

"This value was chosen based on our initial test simulations and inversions (e.g. NCEO (2022)), and the results of previous studies (Jia et al., 2022). In addition, this value proved to be a good compromise in producing simulated column mixing ratios approaching those seen IASI whilst not straying too far from the ICOS-observed values."

Apart from site HTM, variations of simulations with a priori emission rate at other three sites look reasonable considering the coarse resolution of TOMCAT. It also implies that the metrological data and dynamics used in TOMCAT simulations are not main reasons cause large discrepancies. The coarse resolution, as mentioned in the paper, can be one of the important reasons. However, Fig. 6 indicates the negative biases are also very large over land comparing to IASI. Thus, the coarse resolution is not the key reason for such big discrepancies between TOMCAT and IASI.

Yes, we agree that the model's coarse resolution is likely not a major factor in the discrepancies between TOMCAT and IASI. We did not intend to suggest that this was the case in the main text. We do, however, suggest that the coarse resolution may be responsible for some of the discrepancies at the tall tower sites, which the reviewer appears to agree with. We suggest that representation error between the model and the observations at these tall tower sites may be relatively large, especially for a distinct plume of strongly elevated methane concentrations such as in this case. We have clarified this in the main text at the end of the discussion section (Section 5).

4. The results including the IASI observations even became much worse. Followed by the above comment, it would be helpful if the authors can compare simulations with satellite observations at some background sites before the analysis. On one hand, this comparison can be used to check if the system bias exists between model and IASI. On another hand,

the satellite observations can be quite noisy at high latitudes. It also helps to assess the quality of the observations over North Sea.

In response to a comment from the other reviewer, we have updated the ICOS data used for comparison in this paper. Due to this, the posterior comparison at the ICOS site at BIR is greatly improved, indicating that the IASI-based inversion can produce results consistent with the in situ observations. The results are now more consistent between the two sets of observations. We also highlight the good prior and posterior model performance at the UTO site, which is the least affected by the Nord Stream methane plume. The total methane and its variation are both captured well by the model here, leading us to have confidence in its performance regarding the background $CH_4$ values.

To a regional revision by using a transport model like TOMCAT, the boundary condition and initial condition can be quite important. There is little discussion about them, even the spatial distribution of the emission inventory used in this study is not presented and discussed. It is quite difficult to understand why the difference between model simulations with a priori and a posterior inventory only occurs in the dashed black box (plume region) showing in Fig. 6d.

It is correct to say that the initial and boundary conditions are important in this type of inversion. We have now added a figure in the Supplementary material to display the emissions distribution used in these model simulations. The inventories are described in Section 3, where it is noted that the uncertainty contributed by the other emissions should be small in a short run such as this one. We have now also noted the potential effects of their uncertainties at the end of Section 5 as follows:

"The uncertainty induced by the emission inventories should be small compared to the observed plume-related concentrations during a short simulation such as this one, but the initial conditions could introduce biases between the model and satellite. We attempt to account for this through inclusion of the background XCH4 in the inversion state vector, but further investigation into the effect of the initial conditions is warranted."

We have only displayed the results from the simulations in which the plume concentrations, those within the dashed black box, were optimised during the inversion. The background $CH_4$ was more likely to change in the other set of inversions in which the model was optimsed against all XCH4 values.

**Specific comments:**

1, Line 54: It would be better to refer two gas leak points marked with longitude and latitude to red asterisks (i.e., Gas leak 1, 2, 3) in Figure 1.

Agreed, we have added this to the text.

2, Line 124: It would be better to add longitude and latitude along figures. The names of countries are also suggested to be added in Figure 2. The readability can be improved in this way.

We have added longitude and latitude values along the outside of all figure panels. We tried adding country names in figure panels but this decreased the readability of the figures. We did add the country borders, instead.

3, Line 132-134: If I understand correctly, the authors mainly show temporal variations of methane concentrations over North Sea. Are there any observations of IASI over Baltic Sea, where the leakage occurred? If not, the authors should clarify the lack of satellite observations, although IASI retrievals are the only satellite observations that captured a coherent XCH4 enhancement in the days immediately after the leaks began.

IASI was able to retrieve over the Baltic Sea near the source location only on the morning of September 26[th], after which the region was obscured by cloud. To improve clarity, we have adapted this sentence to read as follows:

"The IASI retrievals documented here are the only satellite observations that captured a coherent XCH4 plume from the Nord Stream leaks over the North Sea in the days immediately after the leaks began."

4, Line 193-196: The areas of background in Figure 2 are decided by HYSPLIT model and the uncertainty is estimated by perturbate the area of a black box. However, the number of available observations in each black box is different. How significant is the impact of sampling bias?

The perturbation of the region defined as the 'background' at each overpass affects the derived mean background XCH4 value very little. This indicates that the variance of this value is small across the greater 'background' region - i.e. the area unaffected by Nord Stream but thought to be representative of background XCH4 in the Nord Stream-affected area. We were careful to consider background regions with as much cloud-cleared data as possible for each overpass. We acknowledge that sampling error may play a part in the derived values, but it is difficult to quantify without knowing what XCH4 values were being obscured by cloud. We assume that this sampling error is small due to the low variance produced by the area perturbations. We have added text in Section 3 to clarify this:

"Due to the cloud cover affecting our estimation of the 'background' XCH4 in some cases, it is possible that these estimates include some sampling error that is difficult to quantify due

to the cloud cover itself. We assume that this contribution to the uncertainty is small, however, since perturbing the boundaries of the background region does not affect large changes."

5, Some XCH4 enhancements are quite strong near the western coast of Norway and northern coast of the UK even before 28th. What can be the possible reasons for this?

It is true that there are other large CH4 values prior on September 27[th], particularly near the northern coast of the UK. Whilst observed mixing ratios in this region are not as large as those in the Nord Stream plume, they are elevated in relation to background concentrations. Unfortunately, it is difficult to assess the source of this elevated CH4 as there is no clearly defined nearby event similar to Nord Stream. The TOMCAT model does not replicate these large mixing ratios, so the cause is missing from the bottom-up emissions. Due to the extensive cloud, it is difficult to discern the source region. It is beyond the scope of the paper to define the cause of this CH4, but we do plan to do further grid-scale inversions using this IASI data in the future so might be able to comment at that point. We have commented on this elevated CH4 in the main text, however:

"Large CH4 mixing ratios off the northern coast of the UK on 27th September are likely unrelated to Nord Stream, with their source possibly located across the Atlantic, although this not possible to confirm."

6, Line 236: Why is this emission rate selected? Any background information or explanation?

See above response to main point #3.

---

## Referee Report (RR1)

The authors have clearly addressed each point posted by two reviewers. I recommend it to be published after minor changes.

1, Line 36: I assume that "CH4 mixing ratio" here is "$XCH_4$".
2, Figure 3: Some lines for BIR and HTM are truncated. Why not show the whole range?
3, Line 353: "in inaccurate" should be "inaccurate"
4, Line 420: If I remember correctly, Saunois et al. (2020) built a bottom-up emission inventory at a resolution of 0.1°.
5, Line 430: I don't get the idea why the authors added "These values are close to the value used in the 'modelled' prior, based on the work by Poursanidis et al. (2024)." Does it help to better explain the conclusions than previous version?

---

## Author Response (AR2)

We thank the reviewers for their comments. We hope that we have addressed these concerns appropriately. Our point-by-point response is given below, highlighted as blue text.

**Reviewer 1**

The authors have clearly addressed each point posted by two reviewers. I recommend it to be published after minor changes.
1, Line 36: I assume that "CH4 mixing ratio" here is "XCH4".

No, here we are describing the observed mixing ratios mase at the tall tower measurement sites, so XCH4 is not appropriate in this case.

2, Figure 3: Some lines for BIR and HTM are truncated. Why not show the whole range?

We did consider this option and there are advantages and disadvantages in both cases. We have now updated the BIR panel to show the whole range. However, we prefer to truncate the y-axis at HTM so that the observed variations (black line) can still be clearly seen. This would not be the case if we extended the y-axis further.

3, Line 353: "in inaccurate" should be "inaccurate"

Yes, thank you.

4, Line 420: If I remember correctly, Saunois et al. (2020) built a bottom-up emission inventory at a resolution of 0.1°.
Yes, they record a bottom-up estimate for fossil fuel emissions of $CH_4$ of ~135 Tg yr$^{-1}$. We have adapted the sentence as follows:

"According to Saunois et al. (2020), total global $CH_4$ emissions from fossil fuels amounted to 108 Tg (top-down estimate) or 135 Tg (bottom-up estimate) in the year 2017, approximately 300-370 Gg day$^{-1}$."

5, Line 430: I don't get the idea why the authors added "These values are close to the value used in the 'modelled' prior, based on the work by Poursanidis et al. (2024)." Does it help to better explain the conclusions than previous version?
We have updated this sentence to clarify:

"These values are close to the value used in the 'modelled' prior, based on the work by Poursanidis et al. (2024), indicating that their model provides a good overall estimate of the flux totals."

**Reviewer 2**

I think the authors should specify (e.g. in the introduction) that the NS1 and NS2 pipelines are in fact double pipelines (so four pipelines in total). I do not think this is currently mentioned in the manuscript, however, it should be.
Yes, this is true. We have now included this information on line 46:

"The network is made up of two sets of double pipelines (NS1 and NS2; i.e. four pipelines in total), each originating in Russia and running through the Baltic Sea to Lubmin, Germany (Figure 1)."